# Improvement of the Quality of Life in Aging by Stimulating Autobiographical Memory

**DOI:** 10.3390/jcm10143168

**Published:** 2021-07-18

**Authors:** Alba Villasán Rueda, Antonio Sánchez Cabaco, Manuel Mejía-Ramírez, Susana I. Justo-Henriques, Janessa O. Carvalho

**Affiliations:** 1Faculty of Psychology, Catholic University of Ávila, 05005 Ávila, Spain; 2Faculty of Psychology, Pontifical University of Salamanca, 37002 Salamanca, Spain; asanchezca@upsa.es; 3School of Psychology, CETYS University, Tijuana 22210, Mexico; manuel.mejia@cetys.mx; 4Health Sciences Research Unit: Nursing (UICISA: E), Nursing School of Coimbra (ESEnfC), 3004-011 Coimbra, Portugal; susana.justo.henriques@gmail.com; 5Psychology Department, Bridgewater State University, Bridgewater, MA 02325, USA; janessa.carvalho@bridgew.edu

**Keywords:** aging, quality of life, cognitive stimulation, memory, reminiscence therapy

## Abstract

With notable increases in older adult populations, as well as with the associated cognitive impairments that can accompany aging, there is significant importance in identifying strategies to promote cognitive health. The current study explored the implementation of a positive reminiscence program (REMPOS), a non-pharmacological cognitive therapy that has been previously structured, defined, and tested in a Spanish sample. We sought to improve the quality of life of institutionalized older adults with healthy aging, mild cognitive impairment, and Alzheimer’s disease by utilizing this protocol in these samples. A randomized design with a pre-post measure was conducted over a three-month period. Two types of interventions were used: the experimental groups received REMPOS, and the control groups underwent their regular daily institutional programming with cognitive stimulation techniques. After the intervention, the three experimental groups showed higher cognitive functioning, decreased depressive symptomatology (except for the MCI group) and higher evocation of specific positive memories (except for the MCI group). This study supports the effectiveness of REMPOS and reminiscence therapy, with regard to both cognitive and mood factors in cognitively impaired older adults.

## 1. Introduction

Population aging is exponentially growing and thus we are facing a widespread phenomenon that affects almost all countries with consequences and implications in all areas of life and society [1]. This population growth of people over 60 is notable; it is estimated that by 2050, this growth will double from 11% to 22% [2]. The World Health Organization (WHO) foresees that life expectancy will exceed 90 years by 2030 in some developed countries [3].

There is known deterioration of functions that is associated with degenerative disorders in aging, with consequences for the health and functionality of the elderly [4]. For example, the dementia (or changes in cognition that disrupt functional abilities) prevalence increases exponentially between 65 and 85 years [5]. In 2012, the World Health Organization (WHO) declared dementia as a priority for global public health [6]. While aging is not the cause of dementia, it is one of its greatest risk factors. In a person suffering from dementia, the three areas generally affected are: cognitive, functional, and behavioral, which are closely linked to each other. A careful observation of a person affected by dementia is a great help in identifying what they can and cannot do, and from there, it is possible to develop an intervention program tailored to their needs [7].

Alzheimer’s disease (AD), the most common cause of dementia, affects more than 25 million people worldwide, which incurs a significant costs at the social and health levels [8]. Therefore, it is considered necessary to have an action plan that addresses this pathology in a multidisciplinary way in terms of the form of action and the professionals involved.

Pharmacological therapies (PT) are widely used interventions to combat the progression of AD. However, they have limited effectiveness and they risk adverse side effects, presenting a need to consider a wider range of intervention options, such as non-pharmacological therapies (NPT) [9]. NPTs are defined as non-chemical interventions, theoretically supported, focused and replicable, performed with the patient or the caregiver, that are capable of obtaining relevant benefits such as subjective well-being [10]. The utility of applying NPT in healthy aging and dementias is appealing as it presents without side effects, its implementation is more economical, and it can improve cognition and affect, enhance independence, and increase the quality of life of older adults [11].

One form of NPT is reminiscence therapy (RT). RT is a therapy with elements in common with cognitive stimulation intervention and involves recalling and discussing past activities, events, and experiences, usually using photographs, objects, or music [12]. There are two formats for applying RT: simple reminiscence and life review [12]. Simple reminiscence involves having conversations to stimulating autobiographical memories about various topics from the past, such as holidays, clothes, foods, means of transportation, media, actors, and TV hosts, striking personalities, jobs, and personal experiences. It is an unstructured and spontaneous process that focuses on positive memories, and can be applied in an individual or group format. Life review is more structured and individualized, where the therapist usually guides the individual through significant experiences of their personal lives and trying to make sense of their past experiences. RT is a guided and evaluative process that involves an examination of the patient’s life [13,14].

Various sources support the use of reminiscent narratives (reminiscences of past memories) because they are positive developmental activities that can be individualized and used as a method of cognitive stimulation in older adults, especially when they focus on positive past memories [15,16]. Interventions using reminiscence therapy have become one of the non-pharmacological treatments that has offered proven results of its effectiveness in older adults [8,12,16,17,18,19,20,21]. However, in most studies, the effects of RT for people living with neurocognitive disorders has been shown to have a small positive effect on cognitive function [9,12,22,23]. Regarding specific cognitive domains, there is evidence of improvements in both episodic autobiographical memory and personal semantic memory [9,18,20,21,24]. While much of the literature points to the positive effect of RT on depressive symptoms [9,22,23], other recent studies have found no significant differences [20,21].

The effects of RT on quality of life are inconsistent. Some systematic reviews showing little or no effect [9,12], while others indicate a small or medium effect size on quality of life [23]. In a recent multicenter randomized controlled trial, significant differences were found on quality of life with a small and medium effect size [20,21].

In a previous study, an intervention applying positive reminiscence (REMPOS) therapy [25] in older adults with mild cognitive impairment (MCI) sought to promote the improvement of different psychological processes related to optimal aging [26]. There was a significant increase in cognitive functioning, life satisfaction, self-esteem, and a decrease in depressive symptoms in the experimental group compared to the control group [26]. This supports the effective of REMPOS therapy in a cognitively impaired population.

In the present study, we expand upon the previous study in order to test the effectiveness of the REMPOS program [25] among various cognitive groups. Specifically, the current study compared the intervention across three types of cognitive groups: healthy aging [HA], MCI, and AD to explore its effectiveness on the factors of cognition, affective/mood functioning, and subjective functioning. With relatively limited literature around the effectiveness of REMPOS, the current study adds to the dearth of literature by exploring the effectiveness of this program in a Spanish-speaking population and by exploring its effectiveness in participants with multiple levels of cognitive functioning.

## 2. Method

### 2.1. Participants

The selection of subjects was made in relation to the previously defined sample inclusion criteria: people over 65 years of age, with healthy aging (HA), MCI, and a previous diagnosis of AD who either lived in the residence of, or regularly visited, the day center where the intervention was conducted.

Participants were administered REMPOS intervention (experimental group) or a conventional cognitive stimulation program (control group). We sought to demonstrate cognitive (measured with the Mini Cognitive Examination [MEC] and Montreal Cognitive Assessment [MoCA] instruments), emotional (through the use of the Life Satisfaction Index-Adults [LSI-A] and Geriatric Depression Scale-30 [GDS-30] instruments), and subjective (with the use of the Autobiographical Memory Test [AMT] instruments) changes in our sample through the use of positive reminiscence therapy (REMPOS) through a repeated measures design in the Spanish population. Specifically, we expected significant cognitive improvements in the experimental groups of HA, MCI, and AD, as well as significantly improved emotional variables (depression and life satisfaction) in the experimental groups for HA, MCI, and AD.

### 2.2. Instruments

The instruments used both in the pre-treatment phase (first evaluation), and in the post-treatment phase (second evaluation) were the MoCA [27,28,29], the MEC [30], the GDS-30 [31,32,33], the LSI-A [1,34] and the AMT [35], specific memories for positive stimuli (EPOS), and for negative stimuli (ENEG).

### 2.3. Process

There were six phases within this study. Phase 1 was contact with the institutions. We initially met with the institutions interested in participating in the study, to which we explained the purpose of this study, the planned timing, as well as the methodology to be developed for achievement and development. Once this first contact was made, approximately two or three months before the start of the intervention, the centers agreed to participate in the study.

Phase 2 was the selection of the participants. The sample was selected by screening mass or population (mass screening) incidentally through contact with the day centers and/or residences that agreed to participate in this project. All facilities were located in the city of Salamanca, Spain. The three participating facilities included AFA (Association of Relatives of Alzheimer’s Patients), Residencia Madre de la Veracruz, and Residencia las Mercedarias de la Caridad (in which there was a previous collaboration agreement with the Pontifical University of Salamanca).

Phase 3 was the pre-test and group formation. Each participant was assigned a booklet with the tasks for each session (see next section for the description of the program), furthermore, each task was explained individually and orally.

### 2.4. Description of the Treatment and Control Groups (AD, HA and MCI)

The total sample was divided into six groups: experimental group with three levels (HA, MCI and AD) and the control group with three similar levels (HA, MCI and AD). A total of 77 participants were included, 26 with AD, 24 with MCI and 27 with HA (see Table 1 for details).

Group selection followed predefined criteria. Participants with a previous AD diagnosis became part of that group, all participants in this group participated at a day center. For participants without a previous AD diagnosis or any associated pathology, MEC scores were used to classify them into two possible groups: healthy aging (HA, MEC ≥ 25), and MCI (MEC < 25), in the absence of a specific neuropsychological battery to diagnose MCI or AD in early stages [36]. All participants in the last two groups lived in the residences.

Phase 4 was intervention. Once the pre-test had been carried out, the data obtained were analyzed to form two groups at random. Each group participated in 12 sessions (experimental groups received positive reminiscence (REMPOS) (HA, MCI and AD), while the control groups (HA, MCI and AD) participated in regularly scheduled cognitive stimulation activities through their institutions (detailed in Table 2) occurring simultaneously. The interventions took place twice weekly for one-hour sessions over two months (May–June). In the control groups, attention, perception, memory, language, inhibition, planning, reasoning, calculation, and drawing were the primary focus. In the experimental groups, attention, perception, memory, language, inhibition, planning, reasoning, calculation, drawing, and group dynamics were covered in order to enhance social skills and the expression of positive feelings, as well as to improve the interrelation of the participants.

The guide used for the cognitive stimulation component was developed by the Pontifical University of Salamanca, from which a large part of the exercises that make up this part of the intervention were selected [37]. REMPOS was created by one of the authors of this paper and the protocol is specified in detail elsewhere [25]. Table 2 provides topics of the sessions for both interventions.

Phase 5. Upon completion of the intervention, as assessment of both groups (experimental and controls) was conducted, using the tests specified in phase three (pre-test and group formation). All repeated measurements were made between 3 months and 3.5 months.

## 3. Results

### Data Analyses

Three-factor repeated measures analysis of variance (ANOVA) were run for each dependent variable, each one with the following factors: “time” (pre or post, within subjects), “type of intervention” (experimental or control, between subjects), and “type of aging” (HA, MCI, or AD, between subjects). Results values were considered statistically significant when *p* < 0.05. Statistical analyses were performed using R 4.0.2 [38], rstatix [39], ggplot2 [40], WRS2 [41] and Measure of the Effect (MOTE) [42] packages. When double or triple interactions were significant in the ANOVA, post-hoc analyses were calculated to test the pre-post differences for each type of aging and type of intervention, adjusting *p*-values for multiple comparisons using the Holm correction.

Assumptions checks were run for non-normality using the Shapiro–Wilk test and for the presence of extreme outliers using interquartile ranges. For the MoCA and GDS-30 scores, there were no extreme outliers, and none of the subgroups showed a significant deviance from normality. For MEC scores there were no extreme outliers, and only one out of 12 subgroups (pre-intervention scores for the experimental Alzheimer’s group) showed deviance from normality (*p* = 0.007). MoCA, MEC and GDS-30 scores were analyzed with parametric tests.

The LSI-A, EPOS, and ENEG scores were analyzed with robust and non-parametric tests. For the LSI-A scores, two extreme outliers were identified in the Alzheimer group, with a significant deviance from normality for the pre-intervention scores (*p* = 0.04) of that same group. For the EPOS scores, one extreme outlier was identified in the post-intervention scores of the control Alzheimer group, and three subgroups deviated from normality: post-intervention scores of the control Alzheimer’s group, and both post-intervention scores of the healthy aging groups (*p* = 0.0002, *p* = 0.02, and *p* = 0.0008). The ENEG scores showed at least five extreme outliers, mostly from the pre-intervention scores of the experimental healthy group, and three of the subgroups of the healthy participants deviated from normality (all *p*s < 0.01). Robust ANOVA (based on Wilcox WRS functions) was used for the analyses of the LSI-A, EPOS and ENEG scores, in these cases post-hoc analyses were performed using Wilcoxon paired sample tests.

A total of 77 subjects (58 women) were part of the centers mentioned in the previous section, and residents in the city of Salamanca (Spain). The average age was 83.1 years (mean age men = 79.3 years; women = 84.3 years). Regarding education, 5% had no formal education, 60% had only primary education, 20% finished high school, and 15% completed university. For the main descriptive results of the types of intervention (experimental or control), types of aging (AD, MCI, or HA), and time (pre- or post-intervention), see Table 3.

Results for the MoCA scores showed a significant three-way interaction (F(2,71) = 4.88, *p* = 0.01, η2G = 0.03, 95% CI [0.00, 0.12]) between type of intervention, type of aging, and time. For the AD group, there was a large and significant effect of intervention for experimental group (t(19) = 10.3, *p* < 0.001, Hedges’ g = 2.2, 95% CI [0.46, 3.47]), but not for the control group (t(5) = 0.39, *p* = 0.71, Hedges’ g = 0.13, 95% CI [−0.8, 1]). For the MCI group, there was a significant effect in both the experimental (t(10) = 2.5, *p* = 0.03, Hedges’ g = 0.70, 95% CI [0.03, 1.23]) and the control groups (t(12) = 3.75, *p* = 0.003, Hedges’ g = 0.97, 95% CI [0.35, 1.47]). In the healthy aging group, there was a significant effect in both the experimental (t(13) = 3.06, *p* = 0.009, Hedges’ g = 0.77, 95% CI [0.23, 1.23]), and the control groups (t(12) = 4.49, *p* < 0.001, Hedges’ g = 1.16, 95% CI [0.50, 1.58]). The triple interaction was mostly driven by the fact that both the experimental and control groups in the MCI and HA groups showed similar improvement after intervention, but on the other hand, the experimental AD participants had even higher improvement scores, while the control group showed improvement after intervention (see Figure 1).

Results for the MEC scores also showed a significant three-way interaction (F(2,71) = 3.33, *p* = 0.04, η2G = 0.02, 95% CI [0.00, 0.11]) between type of intervention, type of aging and time. For the Alzheimer’s group, there was a large and significant effect of intervention for the experimental group (t(19) = 4.65, *p* < 0.001, Hedges’ g = 1.0, 95% CI [0.35, 1.50]), and there was also a large and significant effect of time in the control group, but in the opposite direction (t(5) = −2.58, *p* = 0.049, Hedges’ g = 0.89, 95% CI [−0.97, 2.37]), while participants in the experimental group improved their scores after intervention, those in the control group had lower scores on their second assessment. For the MCI group, there was a significant effect in the experimental group (t(10) = 3.73, *p* = 0.004, Hedges’ g = 1.04, 95% CI [−0.02, 1.84]), but not in the control group (t(12) = 2.0, *p* = 0.069, Hedges’ g = 0.52, 95% CI [−0.22, 1.10]). In the healthy aging group, there was also a significant effect in the experimental group (t(13) = 3.61, *p* = 0.003, Hedges’ g = 0.91, 95% CI [0.04, 1.44]), but not in the control group (t(12) = 1.72, *p* = 0.11, Hedges’ g = 0.45, 95% CI [−0.30, 1.05]). Considering the MEC scores, participants in all three cognitive experimental groups showed improvements after intervention. In this case, the triple interaction was driven by the fact that participants with AD in the control group showed significantly lower scores on the second assessment (see Figure 1).

The MoCA and MEC scores showed a strong correlation between them before (r = 0.734, df = 75, *p* < 0.001) and after intervention (r = 0.789, df = 75, *p* < 0.001). For the AD group, the MoCA and MEC scores showed the same clear pattern where only the experimental group benefited from the intervention. The difference found between the MoCA and MEC results in the MCI and HA groups could be interpreted as a matter of difference in sensitivity, where the MoCA group could have been more sensitive to cognitive changes.

For the GDS-30 results, there was no significant three-way interaction (F(2,71) = 0.70, *p* = 0.50, η2G = 0.002, 95% CI [0.00, 0.03]) between type of intervention, type of aging, and time. There was only a significant two-way interaction between type of aging and time (F(2,71) = 8.75, *p* < 0.001, η2G = 0.03, 95% CI [0.00, 0.13]), but not between type of aging and type of intervention (F(2,71) = 1.02, *p* = 0.37, η2G = 0.025, 95% CI [0.00, 0.12]), nor between type of intervention and time (F(1,71) = 0.05, *p* = 0.83, η2G < 0.001, 95% CI [0.00, 0.02]). There were significant main effects of type of aging (F(2,71) = 6.22, *p* = 0.003, η2G = 0.13, 95% CI [0.01, 0.29]), and time (F(1,71) = 19.95, *p* < 0.001, η2G = 0.034, 95% CI [0.00, 0.16]), but not of type of intervention (F(1,71) = 0.38, *p* = 0.54, η2G = 0.005, 95% CI [0.00, 0.08]). In the post-hoc analyses, there was a significant effect of intervention both for experimental (t(13) = 4.78, *p* < 0.001, Hedges’ g = 1.20, 95% CI [0.41, 1.81]) and the control (t(12) = 4.38, *p* < 0.001, Hedges’ g = 1.14, 95% CI [0.26, 1.81]) groups in the healthy aging sample, but only for the experimental group (t(19) = 3.07, *p* = 0.006, Hedges’ g = 0.66, 95% CI [0.12, 1.11]) in the Alzheimer sample. The MCI participants showed no significant difference after intervention in either the experimental (t(10) = 0.36, *p* = 0.728, Hedges’ g = 0.10, 95% CI [−0.54, 0.68]) or the control groups (t(12) = 2.01, *p* = 0.068, Hedges’ g = 0.52, 95% CI [−0.05, 1.04]). As the results (Figure 2) indicate, the two-way interaction found between type of aging and time was driven by an improvement in GDS-30 scores after intervention only in the healthy aging groups, regardless of type of intervention. Healthy participants in the experimental and control groups showed lower GDS-30 scores after intervention.

For the LSI-A results, the robust two-way ANOVA for the difference scores (post minus pre) found a significant main effect of type of aging (*p* = 0.035) and type of intervention (*p* = 0.011), but no interaction between those factors (*p* = 0.083). Post-hoc analyses using Wilcoxon paired sample tests found no significant difference between the post and pre-intervention scores for any subgroup. In general, no effect of the interventions was found on the LSI-A scores (see Figure 2).

For the EPOS results, the robust two-way ANOVA for the difference scores (post–pre) found a significant interaction between type of aging and type of intervention (*p* = 0.002), and a main effect of type of aging (*p* = 0.001), but no main effect of type of intervention (*p* = 0.082). Post-hoc analyses using Wilcoxon paired sample tests found significant differences before and after intervention for the experimental (W = 105, *p* = 0.005) and control (W = 91, *p* = 0.006) groups of healthy participants, and for the experimental group with AD (W = 150, *p* = 0.003) group. Neither the control group with AD nor the experimental or control groups with MCI showed significant differences after the intervention (*p*s > 0.837).

For the ENEG results, the robust two-way ANOVA for the difference scores (post minus pre) found a significant main effect of type of aging (*p* = 0.005), but no significant main effect of type of intervention (*p* = 0.902) nor a significant interaction between type of intervention and type of aging (*p* = 0.372). Post-hoc analyses using Wilcoxon paired sample tests found significant differences before and after intervention for the experimental (W = 88, *p* = 0.015) and control (W = 73, *p* = 0.041) groups of healthy participants, and for the experimental group with Alzheimer’s (W = 80, *p* = 0.049), like the results found in the EPOS scores, but the effect was much smaller in this last group. Neither the control group with Alzheimer’s nor the experimental or control groups with MCI showed significant differences after the intervention (*p*s > 0.993).

For both the EPOS and ENEG scores from the AMT, there was an effect of the intervention for both the control and experimental groups in the healthy aging sample, and only in the experimental group in the Alzheimer’s group, and in all cases scored improved after the interventions (see Figure 3). This pattern is similar to the results for GDS-30. In these affective variables, results suggest that the REMPOS program may have a positive effect in people with Alzheimer’s, but not much effect on people with MCI.

## 4. Discussion and Conclusions

This study explored the cognitive (measured with MEC and MoCA), emotional/affective (measured with LSI-A and GDS-30), and subjective (measured with AMT) functioning of older adults with (AD, MCI) and without (HA) cognitive impairment after positive reminiscence therapy (REMPOS).

The current study explored how positive reminiscence therapy (REMPOS) is related to the well-being of the elderly in the cognitive and affective (mood) variables. There were notably significant improvements between the pretest and posttest scores in relation to the intervention (experimental and controls), in the three cognitive groups (AD, MCI and HA) studied.

While reminiscence was considered as a possible sign of dysfunction and/or deterioration when it occurred at the end of life, it is currently considered to have adaptive functions, serving as a positive predictor of mental health in the elderly [18]. An intervention based on reminiscence therapy was associated with a statistically significant increase in the general cognitive level, a decrease in depressive symptomatology, an increase in life satisfaction, and a greater evocation of specific positive and negative memories [26].

The current study presents with some notable limitations. The sample size is small, namely people with AD in the control group. Our sample groups were defined by a commonly used cognitive screening tool (MEC) with normal group distributions. However, in comparison to another cognitive screening tool (MoCA), our group distributions did not fit typical score cutoffs even when the correlation between those measures was high. There are several considerations, among which are education, which can have notable effects on MoCA performance. Specifically, the majority of our sample had education levels at elementary school or below, which could have had performance effects on this measure. Among other considerations, all participants in groups defined using MEC scores lived in geriatric residencies, which could be associated with a higher prevalence in cognitive decline still undiagnosed or in early stages, or with a context lacking stimulating activities. Thus, while we feel the groups were appropriately identified, they did not perform as traditionally as we expected on all cognitive tasks. Nonetheless, all three of our experimental groups did show improvements in this task.

To highlight some of the positive effects, we underscore the reduced depressive symptomatology in two experimental groups (AD and HA), compared to the control groups (MCI, AD and HA). The question remains as to why our MCI experimental group did not respond as strongly to the intervention as the other experimental groups. Possible considerations are the aforementioned group factors such as limited education and atypically baseline group measurements; that is, it is possible that at least some participants in the MCI group may have been better characterized as AD. Another possible consideration is the degree of awareness or agnosia that is specific to the MCI population; this very specific level of awareness in this group may have affected their confidence in the intervention and thus affected their outcomes. Notably, there have been some reflections on the variability of MCI groups in intervention outcomes and how it can affect effectiveness results [43,44].

Our overall results are consistent with the results of various studies that propose the reduction of depressed moods as one of the main objectives of this type of therapy in older adults [17,45,46,47]. Any intervention, particularly non-pharmacological, that can reduce depressive symptoms to any extent in older adults has clinical importance [48], as depressive symptoms are quite common in aging. This result is consistent with previous studies that show that reminiscence plays an important role in the health of the elderly because of its therapeutic and adaptive nature, and therefore, influences their quality of life [8,48]. This study allows us to approach the improvement of the quality of life of the elderly from a non-pharmacological perspective. While reminiscence therapies have been showing their effectiveness in previous studies, the current study goes a step further by doing it with older adults of different cognitive abilities.

We can conclude that this work shows that positive results were obtained from the application of an NPT program in aging. It is important to highlight that individuals with cognitive impairment benefit more from it, although its widespread application seems to be optimal in relation to cognitive and emotional aspects. Overall, the results found are quite encouraging in order to indicate the need to continue to promote studies on this topic, as the use of positive reminiscence therapy has demonstrated a significant decrease in depressive symptomatology, an increase in cognitive level, life satisfaction, and the evocation of specific memories, all imperative factors in the psychological well-being and quality of life of the elderly.

## Figures and Tables

**Figure 1 jcm-10-03168-f001:**
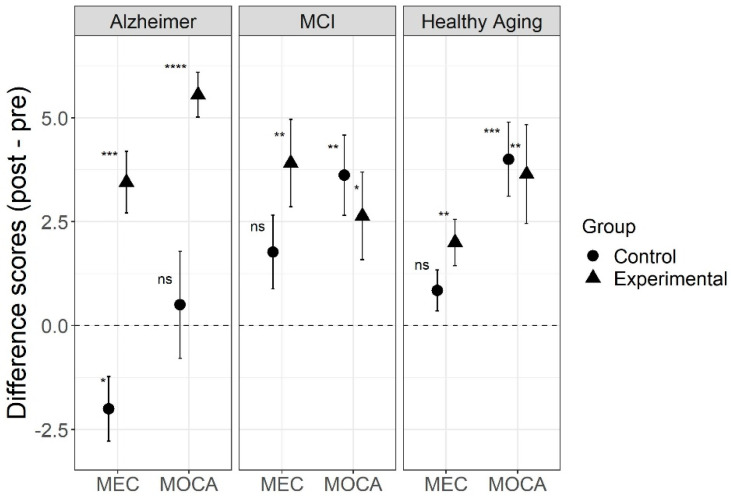
MoCA and MEC difference scores for three types of aging before and after intervention, comparing the REMPOS program (experimental group) to standard cognitive stimulation (control group). Note: Error bars indicate standard errors. Alzheimer = Alzheimer’s disease; MCI = mild cognitive impairment; MoCA = Montreal cognitive assessment; MEC = mini cognitive examination. * *p* < 0.05, ** *p* < 0.01, *** *p* < 0.001, **** *p* < 0.0001, testing a significant difference from zero.

**Figure 2 jcm-10-03168-f002:**
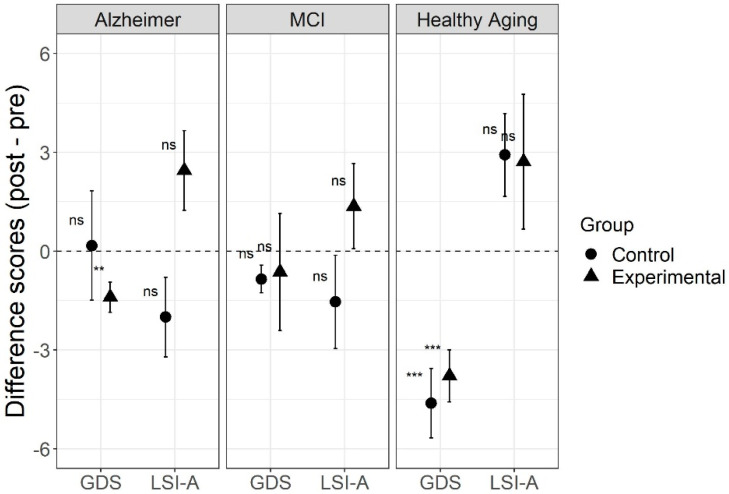
Geriatric depression scale-30 (GDS-30) and life satisfaction index-adults (LSI-A) scores for three types of aging before and after intervention, comparing the REMPOS program (experimental group) to standard cognitive stimulation (control group). Note: Error bars indicate standard errors. Alzheimer = Alzheimer’s disease; MCI = mild cognitive impairment; GDS = geriatric depression scale-30; LSI-A = life satisfaction index-adults. ** *p* < 0.01, *** *p* < 0.001, testing a significant difference from zero.

**Figure 3 jcm-10-03168-f003:**
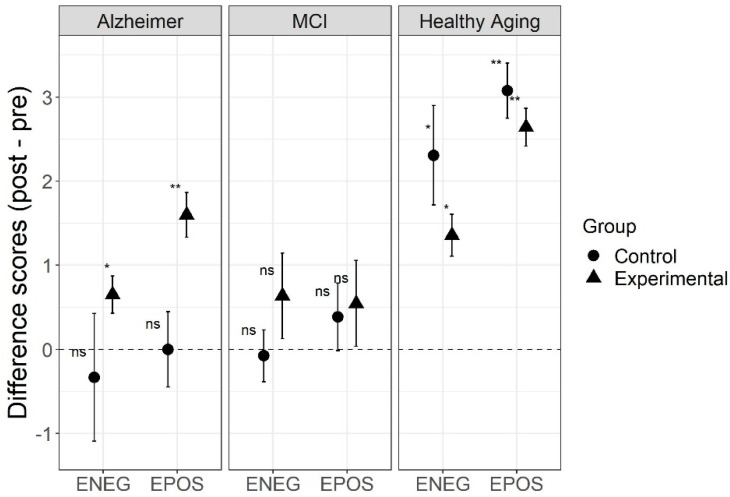
Autobiographical memory test (AMT) scores for three types of aging before and after intervention, comparing the REMPOS program (experimental group) to standard cognitive stimulation (control group). Note: Error bars indicate standard errors. Alzheimer = Alzheimer’s disease; MCI = mild cognitive impairment; EPOS = autobiographical memory test, specific memories for positive stimuli; ENEG = autobiographical memory test, specific memories for negative stimuli. * *p* < 0.05, ** *p* < 0.01, testing a significant difference from zero.

**Table 1 jcm-10-03168-t001:** Group formation.

Groups	AD	MCI	HA	Total
Control	6	13	13	32
Experimental	20	11	14	45
Total	26	24	27	77

Abbreviations: AD = Alzheimer’s disease; MCI = Mild cognitive impairment; HA = Healthy aging.

**Table 2 jcm-10-03168-t002:** Themes of interventions.

Session	Positive Reminiscence	Cognitive Stimulation
1	Introduction to reminiscence	Keys to optimize registration: concentration
2	Things of everyday life	Organization of information
3	I present-past-future	Display and erroneous attributions
4	Relationships	Importance of language
5	Important dates	Routes and semantic knowledge
6	Popular parties	Understanding of texts and procedural knowledge
7	Work and labor	Calculation and arithmetic
8	Games	Calculation capacity development
9	Remembering loved ones	Relational memory training I
10	Music and memories	Relational memory training II
11	Reirpos (positive emotions through laughter)	Importance of care and self-regulation
12	Laugh more live more	Breathe

**Table 3 jcm-10-03168-t003:** Descriptive statistics for the six main cognitive variables assessed before and after intervention (control or experimental groups) for the three types of aging groups. Means (SD).

	Control		Experimental	
	Pre	Post	Pre	Post
**AD group**	*n* = 6		*n* = 20	
MoCA	19.2 (5.23)	19.7 (2.58)	11.7 (3.21)	17.2 (3.32)
MEC	28 (1.90)	26 (2.45)	20.4 (3.22)	23.9 (2.77)
GDS-30	7.67 (2.16)	7.83 (4.62)	9.2 (4.20)	7.8 (4.03)
LSI-A	27 (6.51)	25 (7.69)	24.6 (5.46)	27.0 (5.84)
AMT-EPOS	2.17 (0.75)	2.17 (0.41)	1.9 (0.97)	3.5 (0.89)
AMT-ENEG	2.5 (1.23)	2.17 (0.75)	2.15 (1.09)	2.8 (0.89)
**MCI group**	*n* = 13		*n* = 11	
MoCA	11.2 (4.25)	14.8 (3.36)	13.1 (4.59)	15.7 (5.08)
MEC	19.9 (3.45)	21.7 (3.40)	19.6 (3.98)	23.5 (4.53)
GDS-30	13.5 (5.55)	12.6 (6.53)	13.4 (5.57)	12.7 (6.40)
LSI-A	22.6 (6.5)	21.1 (6.75)	20.4 (4.41)	21.7 (6.25)
AMT-EPOS	1.23 (1.17)	1.62 (0.77)	1.36 (1.12)	1.91 (1.14)
AMT-ENEG	2.46 (0.88)	2.38 (1.04)	1.73 (1.42)	2.36 (1.21)
**HA group**	*n* = 13		*n* = 14	
MoCA	17.6 (3.62)	21.6 (2.53)	20.7 (4.27)	24.4 (3.15)
MEC	26.4 (1.39)	27.2 (2.24)	28.6 (2.53)	30.6 (2.85)
GDS-30	15.1 (4.70)	10.5 (3.53)	11.9 (5.10)	8.07 (3.91)
LSI-A	18.5 (5.88)	21.5 (3.69)	22.4 (5.26)	25.1 (5.01)
AMT-EPOS	1.23 (0.93)	4.31 (0.95)	1.79 (0.80)	4.43 (1.02)
AMT-ENEG	1.62 (1.50)	3.92 (1.60)	2.86 (0.54)	4.21 (0.98)

Abbreviations: AD = Alzheimer’s disease; AMT-EPOS = autobiographical memory test, specific memories for positive stimuli; AMT-ENEG = autobiographical memory test, specific memories for negative stimuli; MCI = mild cognitive impairment; HA = healthy aging; MoCA = Montreal cognitive assessment; MEC = mini cognitive examination; GDS-30 = geriatric depression scale-30; LSI-A = life satisfaction index-adults.

## Data Availability

The original contributions presented in the study are included in the article further inquiries can be directed to the corresponding author.

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
