# Peer review of "Improvement of the Quality of Life in Aging by Stimulating Autobiographical Memory"

_jcm, 2021, doi:10.3390/jcm10143168_

Round 1
Reviewer 1 Report
This study showed positive effects of a reminiscence therapy on three groups of older adults with differing levels of cognitive function/health.
It is good to see demonstrations of effective interventions on older adult samples, especially dementia groups. I was not too sure about how this study differs from existing reminiscence work based on the information provided. I feel the research definitely contributes to the literature but it may not have sufficient theoretical novelty for this journal.
Major issues
Further justification of the novelty of this study may be required.
The importance of control task is crucial in these types of studies. More details of what the control group were doing would be good, specifically what tests. As far as I understand it, the experimental group also did some of the control tasks (lines 140-145) – did the experimental group receive more attention (i.e., control tasks plus REMPOS)?
Minor issue
It would have been nice to have verbal summaries of interactions rather than lists of significant and non-significant post hoc effects, although the graphs were good so this is not a huge problem
Line 327 may be a typo otherwise sentence needs revising as I could not follow it.
Author Response
Thank you for your thoughtful edits and comments on our manuscript submitted to the Journal of Clinical Medicine: “Improvement of the quality of life in aging by stimulating autobiographical memory.” We appreciate the time from reviewers providing helpful comments to help improve our manuscript. We outlined below our responses to their comments and the changes made to the manuscript; changes to the manuscript are provided using track changes. We hope the revisions will make the paper acceptable for publication.
Further justification of the novelty of this study may be required.
ANSWER 1: We have edited the Introduction to provide more details around the novelty around our specific study relative to the research conducted on this topic (lines 70-126).
The importance of control task is crucial in these types of studies. More details of what the control group were doing would be good, specifically what tests. As far as I understand it, the experimental group also did some of the control tasks (lines 140-145) – did the experimental group receive more attention (i.e., control tasks plus REMPOS)?
ANSWER 2: We thank the reviewer for raising this important point. We have edited the Method section to more specifically refer to Table 2 which provides more details around the control tasks in comparison to the experimental tasks. Briefly, the control tasks were active cognitive tasks as provided by the regular care at their institution.
Around where Table is referred to, I’d suggest adding a mention to “cognitive stimulation activities” when mentioning the control group:
Line 173: “... and control groups (...) participated in regularly scheduled cognitive stimulation activities through...”
It would have been nice to have verbal summaries of interactions rather than lists of significant and non-significant post hoc effects, although the graphs were good so this is not a huge problem
ANSWER 3: Thank you for raising this point. We have added a verbal narrative to our interactions on lines 269 and 295.
Line 327 may be a typo otherwise sentence needs revising as I could not follow it.
ANSWER 4: Correction done. Deleted “such as specifically develop”.
Again, we thank the reviewers for their suggestions and hope that our edits satisfactorily address the reviewers’ concerns.

Reviewer 2 Report
The authors present an interesting study into the effectiveness of a positive reminiscence therapy in comparison with a regular cognitive stimulation program in three different groups of older adults. The study appears to be well executed, but a number of issues in the manuscript need clarification.
- In the abstract the authors state that they used a randomized design. However, in the manuscript there is no mention whatsoever as to how the randomization was performed.
- The manuscript refers to other publications for a description of REMPOS. However, these publications are in Spanish. A more explicit description is welcome for readers in English. From the method section, phase 4, I conclude that the REMPOS program differs from the regular cognitive stimulation program in the control group only with the aspect of group dynamics. If this is the distinctive feature of REMPOS it needs to be elaborated. From the title of the manuscript, I expected that stimulating autobiographical memory would be the essential feature. So, the specific additional feature of the REMPOS program is still unclear. In the discussion the specific benefit of the program in comparison with a conventional cognitive stimulation program should be discussed as well.
- In the last sentence of the introduction the authors write: “the current study compared 80 the intervention across three types of cognitive groups, healthy aging [HA], MCI, and 81 AD to explore its effectiveness on factors of cognition, mood, and quality of life”. But further they distinguish cognitive, emotional and subjective functioning. It is helpful to use the same terminology throughout the manuscript.
- In line 95-98 the authors write up their hypotheses where they expect improvements in the experimental group. But with a design such as this I would expect more explicit expectations of the comparison with the control group.
- In line 97: “... and significantly increased in the experimental 97 groups for HA, MCI, and AD” between significantly and increased one or more words are missing (subjective functioning?).
- In the description of treatment and control groups the sentence (line 125-126): “The groups included 13 in the HA and MCI groups and 11 people in the AD groups.” the numbers are not in line with table 1.
- The inclusion of 6 people with AD in the control group is small. This may need attention in the limitations.
- In table 2 under session 11 the word Reirpos is not clear to me.
- In the results section (line 174) the abbreviations EPOS and ENEG suddenly appear. In table 3 these are explained, but it is helpful for the reader to mention these two aspects of autobiographical memory in the instruments paragraph. Also, in table 3 TMA is used, where in the heading of figure 3 AMT is used. It is advised to be consistent.
Author Response
Thank you for your thoughtful edits and comments on our manuscript submitted to the Journal of Clinical Medicine: “Improvement of the quality of life in aging by stimulating autobiographical memory.” We appreciate the time from reviewers providing helpful comments to help improve our manuscript. We outlined below our responses to their comments and the changes made to the manuscript; changes to the manuscript are provided using track changes. We hope the revisions will make the paper acceptable for publication.
How the authors explain the fact that MCI group does not respond the stimulation? It should be addressed in the discussion
ANSWER: The reviewer raises an important point that we now address in the Discussion, devoting lines 403-412 to possible considerations as to this group difference.
In addition, a strength of this study has been to have three groups, with the same intervention differentiating cognitive levels) and the clarification of differentiated effects depending on the diagnoses (greater or lesser involvement). This key is a novelty in relation to the inconsistency of results in previous studies that have manipulated the methodologies but have not considered the level of cognitive impairment.
One limitation of the study is that we did not record or control negative life events or other events that could potentially impact higher depression scores, which should be considered in future research. Controlling this variable would clarify the important question of whether reminiscence programs prevent depression in the future, in addition to reducing depression during the intervention.
Again, we thank the reviewers for their suggestions and hope that our edits satisfactorily address the reviewers’ concerns.

Reviewer 3 Report
In this paper, Villasan Rueda et al tested the hypothesis that REMPOS, reminiscence therapy, could improve both cognitive and mood factors in cognitively impaired older adults .They compared the intervention across three types of cognitive groups, healthy aging, exploring the effectiveness of REMPOS on factors of cognition, mood, and quality of life. They report significant improvements between the pretest-posttest scores in relation to the groups in the group of AD patients, healthy aging partecipants but not in the MCI group
This is a nice and well-conducted study with a good design but a small sample size. The methods are clearly spelled out and the results are presented effectively. The topic is of great importance and emphasize the role of non-pharmacological intervention in dementia
I have a comment:
How the authors explain the fact that MCI group does not respond the stimulation? It should be addressed in the discussion
Author Response
Thank you for your thoughtful edits and comments on our manuscript submitted to the Journal of Clinical Medicine: “Improvement of the quality of life in aging by stimulating autobiographical memory.” We appreciate the time from reviewers providing helpful comments to help improve our manuscript. We outlined below our responses to their comments and the changes made to the manuscript; changes to the manuscript are provided using track changes. We hope the revisions will make the paper acceptable for publication.
In the abstract the authors state that they used a randomized design. However, in the manuscript there is no mention whatsoever as to how the randomization was performed.
ANSWER 1:Effectively the reviewer is right. The paragraph on “Phase 4. Intervention” has been amended.
The manuscript refers to other publications for a description of REMPOS. However, these publications are in Spanish. A more explicit description is welcome for readers in English. From the method section, phase 4, I conclude that the REMPOS program differs from the regular cognitive stimulation program in the control group only with the aspect of group dynamics. If this is the distinctive feature of REMPOS it needs to be elaborated. From the title of the manuscript, I expected that stimulating autobiographical memory would be the essential feature. So, the specific additional feature of the REMPOS program is still unclear. In the discussion the specific benefit of the program in comparison with a conventional cognitive stimulation program should be discussed as well.
ANSWER 2: The text (line 177 and following) has been modified with the clarification requested by the reviewer:
"In the experimental groups, attention, perception, memory, language, inhibition, planning, reasoning, calculation, drawing and group dynamics were covered in order to enhance social skills, the expression of positive feelings and improve the interrelation of the participants ”.
The description of the activities of the control group has been clarified with the following explanation:
“… In the experimental groups the same cognitive processes were addressed, but for this group the activities were designed differently, so the focus was not on improving these cognitive processes (registration, retention and retrieval of information), but on improving social skills, the expression of positive feelings and improving the interrelation between the participants where group dynamics was also focused (see Table 2 for topics of each type of intervention)”.
In the last sentence of the introduction the authors write: “the current study compared the intervention across three types of cognitive groups, healthy aging [HA], MCI, and AD to explore its effectiveness on factors of cognition, mood, and quality of life”. But further they distinguish cognitive, emotional and subjective functioning. It is helpful to use the same terminology throughout the manuscript.
ANSWER 3: We appreciate this point and recognize the importance of consistent terminology. As such, the Introduction now refers to the original domains (line 125), which has been clarified in the Method (lines 136-140)
In line 95-98 the authors write up their hypotheses where they expect improvements in the experimental group. But with a design such as this I would expect more explicit expectations of the comparison with the control group.
ANSWER 4: The hypotheses are modified as follows:
“Specifically, we expected significant and higher cognitive and emotional improvements (depression and satisfaction with life) in the experimental groups than in the control groups of each type of cognitive level (HA, MCI and AD), therefore, we expected interactions significant between time (measures before and after the intervention) and experimental condition (experimental or control groups) for each cognitive and affective measure”.
In line 97: “... and significantly increased in the experimental groups for HA, MCI, and AD” between significantly and increased one or more words are missing (subjective functioning?).
ANSWER 5: Sentence altered.
In the description of treatment and control groups the sentence (line 125-126): “The groups included 13 in the HA and MCI groups and 11 people in the AD groups.” the numbers are not in line with table 1.
ANSWER 6: The entire sentence has to be changed. Instead of the sentence, it could read:
“A total of 77 participants were included, 26 with AD, 25 with MCI and 27 with HA (see Table 1 for details).”
But also the table could be clearer, Just changing the first data row to be the last, and changing the label to “Total”.
Table 1.
Group formation.
|
Groups |
AD |
MCI |
HA |
Total |
|
Control |
6 |
13 |
13 |
32 |
|
Experimental |
20 |
11 |
14 |
45 |
|
Total |
26 |
24 |
27 |
77 |
Abbreviations: AD = Alzheimer’s disease; MCI = Mild cognitive impairment; HA = Healthy aging.
The inclusion of 6 people with AD in the control group is small. This may need attention in the limitations.
ANSWER 7: We specify this limitation in the conclusions.
In table 2 under session 11 the word Reirpos is not clear to me.
ANSWER 7: "Reirpos" (positive emotions through laughter).
In the results section (line 174) the abbreviations EPOS and ENEG suddenly appear. In table 3 these are explained, but it is helpful for the reader to mention these two aspects of autobiographical memory in the instruments paragraph. Also, in table 3 TMA is used, where in the heading of figure 3 AMT is used. It is advised to be consistent.
ANSWER 8: The correct abbreviation for Autobiographical Memory Test is AMT. Table 3 changed. Changed all the text where it was TMA by AMT.
Again, we thank the reviewers for their suggestions and hope that our edits satisfactorily address the reviewers’ concerns.

Round 2
Reviewer 1 Report
The study has been conducted well but could be more novel or more theoretically motivated. It adds the the existing literature in this field by providing further evidence of positive outcomes from reminiscence therapy.
Reviewer 2 Report
The authors have adequately replied to my comments and the manuscript has improved considerably.